# Anti-Screenshot Watermarking Algorithm for Archival Image Based on Deep Learning Model

**DOI:** 10.3390/e25020288

**Published:** 2023-02-03

**Authors:** Wei Gu, Ching-Chun Chang, Yu Bai, Yunyuan Fan, Liang Tao, Li Li

**Affiliations:** 1School of Computer Science and Technology, Anhui University, Hefei 230039, China; 2Department of Computer Science, University of Warwick, Coventry CV47AL, UK; 3School of Computer Science and Technology, Hangzhou Dianzi University, Hangzhou 310018, China

**Keywords:** image watermarking, anti-screenshot, Stegastamp, DLM, archival image

## Abstract

Over recent years, there are an increasing number of incidents in which archival images have been ripped. Leak tracking is one of the key problems for anti-screenshot digital watermarking of archival images. Most of the existing algorithms suffer from low detection rate of watermark, because the archival images have a single texture. In this paper, we propose an anti-screenshot watermarking algorithm for archival images based on Deep Learning Model (DLM). At present, screenshot image watermarking algorithms based on DLM can resist screenshot attacks. However, if these algorithms are applied on archival images, the bit error rate (BER) of the image watermark will increase dramatically. Archival images are ubiquitous, so in order to improve the robustness of archival image anti-screenshot, we propose a screenshot DLM “ScreenNet”. It aims to enhance the background and enrich the texture with style transfer. Firstly, a preprocessing process based on style transfer is added before the insertion of an archival image into the encoder to reduce the influence of the screenshot process of the cover image. Secondly, the ripped images are usually moiréd, so we generate a database of ripped archival images with moiréd by means of moiréd networks. Finally, the watermark information is encoded/decoded through the improved ScreenNet model using the ripped archive database as the noise layer. The experiments prove that the proposed algorithm is able to resist anti-screenshot attacks and achieves the ability to detect watermark information to leak the trace of ripped images.

## 1. Introduction

Archival images have become one of the most important documentary references in the history of modern social development. It is the valuable historical image directly formed by state institutions, social organizations and individuals during their social activities. The archival images play an unique and valuable role for both individuals and organizations. It is an information link between the past and the future that perpetuates human memory. By recording people’s experiences, their development history can be preserved intact. Archival images play an important role in the history of human development, while actively promoting the advancement of individuals. The institutions that provide the service of keeping archival images are the local talent market, each district and county talent market and street office, etc. According to laws and regulations, archival images belong to the national statutory, mandatory and vested public information. No individual may retain and destroy these images.

Moreover, archival images record the main experience, political and ideological style of each person and serve as a proof, basis and reference. For the employer, the Human Resources (HR) department can use archival images to evaluate employees and select talents. It provides an important basis for the salary setting and employee promotion of the employer. The individual appraisal, reward and punishment of the work unit are definitely put into the archival images. Therefore, personnel management can work better in a uniform way. Some workplaces are required to provide archival images for recruitment. This can be a serious problem if the copyright of archival images is not well-protected.

Image watermarking algorithms should have strong robustness because the watermarked image may be attacked in the transmission process. Common attacks include cropping, compression, Gaussian noise, etc., all of which may change the pixel value of the watermark image, and then the watermark information cannot be extracted smoothly. Screenshot attack is a very complex image attack that includes perspective deformation and a moiré pattern. In this paper, the noise between the screen and the camera exists commonly. However, the traditional robust watermarking algorithms show low robustness, so it is necessary to introduce a DLM to improve the performance of the algorithm.

Recently, DLMs such as Stegastamp [1] have been widely applied in computer vision, natural language processing and other fields. DLMs have achieved excellent results in experiments. Therefore, researchers also explore image watermarking algorithm based on DLM. Zhu et al. [2] initiated the HiDDeN model, which introduces a variety of noises between the encoder and decoder to improve the robustness of the auto-encoder. However, since the noise layer only considers Gaussian blur, cropping, JPEG compression and other attacks, it cannot effectively resist the problems of a moiré attack, perspective deformation and color transformation. In order to simulate screenshot attacks in real scenes, Wengwowski et al. [3] proposed the light field messaging (LFM) system. LFM specially trains a network to simulate the effect of cover image to screenshot process, so a dataset of 1,000,000 images is established. However, the dataset takes up a lot of memory and it is too difficult to obtain. Moreover, LFM has a good effect on screenshot attacks, and it has limitations for print attacks.

Many researchers apply the DLM directly as the encoder or decoder. Fang et al. [4] generated several centrosymmetric Gaussian noises on the red channel of the color image to determine the embedding position of the watermark information. They trained a decoder with DLM to extract the watermark on the blue channel of the color image. To resist the screenshot attack, Fang et al. [4] designed an enhancing subnet to improve the accuracy of watermark extraction. In a subsequent work, they applied the anti-screenshot DLM to dynamic images [5]. This algorithm embeds watermark information in the red or blue channel of the image in turn. Due to the frequent switching of the image in a short time, the color interchanges make the watermark information undetectable to the human eye. Fang et al. [5] proposed DLM-based decoder with attention mechanism to improve the accuracy of the model. Consequently, their algorithm has significantly improved in imperceptibility and robustness compared with [4].

DLM can be used not only to embed or extract watermark information directly, it can also be used to assist in image watermark extraction. Li et al. [6] used DLM for finding feature points that can resist screenshot attacks. The watermark information is embedded in the region around the feature points, so feature point localization has a very important position in watermark extraction. DLM can find the feature points that contain the image watermark in the case of screenshot attack on the image. Additionally, Li et al. [7] added an anti-screenshot noise layer to the end-to-end DLM. The noise layer is presented between the encoder and decoder and used to simulate possible attacks on watermarked images. This increases the training time of the DLM to some extent, but the robustness of the anti-screenshot watermarking algorithm is significantly improved. This ensures that when a screenshot is taken of a watermarked image, the decoder can still extract the watermark information accurately.

Matthew et al. [1] proposed the Stegastamp model, in which the differentiable image perturbations were applied between the encoder and decoder to simulate the distortion caused by print attacks and screenshot attacks. The excellent watermarking algorithm is robust to both screen attack and print attack, but the Stegastamp model is low robustness against the cropping attack. Therefore, the image cropping attack has limitations, which leads to the poor quality of a dense image.

In this paper, we propose a robust archival image watermarking algorithm based on the Stegastamp model. The proposed algorithm structure includes five parts: the preprocessing network adds color and texture into the archival image. The encoder embeds the watermark information into the cover image successfully. It also minimizes the difference between the watermarked image and the cover image, and finally generates the watermarked image with good visual quality. The noise layer of Stegastamp is designed to distribute the watermarked image generated by the encoder to simulate the effect of screenshot attacks. The purpose of the moiré network is to further add real moiré to the distributed watermarked image. The decoder is to extract watermark information from the moiré-watermarked image. The main contributions of this paper are as follows:A style transfer network is applied to improve the richness of background texture for archival images. So, the robustness of the image watermarking algorithm can be maintained.A moiré network is constructed to produce the moiré archival image dataset. In addition, moiré images taken in realistic environments are added to build a rich database of screenshot images.The anti-screenshot capability of the Stegastamp model is improved by placing these datasets in the noise layer. It solves the problem of a lack of standard moiré dataset in digital watermarking field.

## 2. Related Works

In this section, we introduce the Stegastamp model and style transfer in details.

### 2.1. Stegastamp

Recently, plenty of learning-based algorithms leveraging encoder and decoder to embed different kinds of watermark in images have been proposed, such as the Stegastamp. The Stegastamp applies DLM to the screenshot watermarking algorithm by modeling more realistic distortions of watermarked image. Firstly, the cover image and the expanded watermarked sequence are fed into the encoder. Then, the watermarked image output from the encoder goes through the printing and screenshot process. To improve the robustness of the algorithm, during the model training, watermarked image is added with some attacks by the noise layer, such as perspective warp, motion/defocus blur, color manipulation, noise and JPEG compression. The noise layer makes the watermarked image closer to the real application scenario. Therefore, the decoder can robustly decode the watermarked image even during the screenshot process. The decoder locates, corrects and extracts the watermarked image in the captured image to form the corrected image. Finally, the decoder extracts the watermark information from the corrected image.

Stegastamp mainly improves the robustness of the image watermarking algorithm against moiré attacks, which are interference streaks that appear on surface of the photographed object. For example, during screenshot, the image displayed on the shooting screen will appear as a moiré pattern.

However, the Stegastamp model is vulnerable to visible noise on the watermarked image during the encoding process, which very unfavorable for the archival images with a single color background, so that it reduces the robustness and imperceptibility of the watermarking algorithm. Some researchers have refined Stegastamp in the past two years; however, the performance of the model is still not well suited for archival images [8,9]. Therefore, the performance of the StegaStamp model for archival images needs to be improved.

### 2.2. Style Transfer

Style transfer is to remain the content of one image unchanged, but its style is changed to another image. Style transfer is often applied to simulate the creative style of artists who have passed away. Due to the excellent performance of DLM, it has been used to accomplish research work on style transfer [10,11,12,13]. For example, Chen et al. [10] proposed a DLM for cartoon style transfer (CartoonGAN). This Generative Adversarial Network (GAN) is well able to migrate the style of images to cartoon style. Li et al. [11] proposed CariGAN to generate caricatures. Their model ensures that the face of the cartoon characters does not show unsightly deformations and generates more details in key areas. Style transfer is not limited to image processing; it can also be applied to automatically generate fonts. Zhao et al. [12] proposed CycleGAN for both text and image. They introduced a new loss function to train the DLM, which better guarantees the style of the synthesized images. In order to apply style transfer in real scenarios, Zhang et al. [13] proposed CSST-Net for tile watermarking algorithm, which has a good performance in watermarking capacity and robustness in addition to good style transfer effect.

Archival images include a large number of white areas, so they are very unsuitable for embedding watermark information directly. Therefore, in this paper, we propose to apply the VGG [14] model to perform style transfer on archival images to improve the imperceptibility of the watermarking algorithm.

## 3. Proposed Algorithm

The flow diagram of the proposed robust watermarking algorithm is shown in Figure 1. Firstly, the cover image is preprocessed. The watermark information with size 100 bits is fully connected to obtain the tensor. Then, the tensor is upsampled to make the size consistent with the image size after preprocessing. After a series of convolution operations, the watermark is embedded into watermarked image. Then, the noise layer simulates the screenshot attack in a real scene with watermarked image. After the watermarked image with noise is attacked by real moiré, the decoder extracts the watermark information through a series of convolution operations and fully connects.

### 3.1. Archival Image Preprocessing

Style transfer allows us to modify the style of an image to another style without changing the content of the image. This is great for archival images where content needs to be protected. More importantly, style transfer makes the background of the archive image more suitable for embedding watermark information. An archival image has a large range of single background. However, the watermarking in a single background will be very conspicuous, which leads to terrible algorithmic imperceptibility. Therefore, the archival image needs to be preprocessed before being inserted into the model. In this work, we apply the style transfer algorithm to preprocess the archival image. This causes the archival image to add different colors and textures, which is more conducive to embedding an image watermark, improving both the imperceptibility and robustness of the proposed algorithm. The structure of the preprocessed archival image is shown in Figure 2.

The preprocessing model applies the trained VGG network [14] as the encoder to extract features. The structures of the transfer encoder and loss encoder are identical. These encoders cover and output the features of different levels of convolution layer by jumping connection. Then, adaptive instance normalization (AdaIN) is applied to adjust the first-order and second-order statistics of features. The spatial mask automatically adjusts the level of stylization. The structure of the decoder is basically symmetrical to the encoder, and it needs to be trained from the beginning. The discriminator is used to predict the style types of the image and distinguish the authenticity of the transfer image at the same time.

### 3.2. Encoder Construction

Encoder’s architecture is similar to the U-net [15]. The encoder is mainly applied to embed the watermark information into the transfer image and minimize the difference between the watermarked image and the transfer image. It concatenates the preprocessed cover image and watermark information to form a tensor. To balance the watermark capacity and the image quality of the generated watermark-containing images, a compromise is adopted to encode 100 bits of watermark information uniformly on a 400 × 400 × 3 carrier image. Figure 3 shows the structure of Encoder.

Before inputting the carrier image and watermark information into the encoder, some pre-processing is needed. First, the 100-bit watermark information is fully concatenated to obtain a 7500-dimensional tensor, and the dimension is modified to obtain a 50 × 50 × 3 tensor, and then the tensor is upsampled to make the dimension of the tensor consistent with that of the carrier image to obtain a 400 × 400 × 3 tensor. The tensor is then cascaded with the carrier image to obtain a 400 × 400 × 6 tensor. The tensor is fed into the encoder for a series of convolution operations. During the convolution process, the high level feature map is upsampled. So, the size of the upsampled high level feature map and its corresponding low level feature map are the same. Finally, the high-level feature map and the low-level feature map are cascaded (concatenate) by channel dimension, so that the low level features can be preserved.

The tensor undergoes a series of convolution, pooling, and downsampling operations, and then combines the previously extracted features for upsampling. Conv_1~Conv_10 represent the convolution operations, using 3 × 3 convolution kernels with the field of perception, the step size of Conv_1 and Conv_6~Conv_10 are set to 1, the step size of other convolutions is set to 2, and the step size of Conv_11 is set to 1. Up_6~Up_9 represent the upsampling operations, using 2 × 2 convolution kernels with the step size set to 1. The activation functions are all ReLU functions, the fill type is “same”, and the initializer of the convolution kernel is he_normal. The cascade operation is performed in the channel dimension. Finally, when the loss function reaches the standard, the encoder generates the watermarked image with good visual quality.

### 3.3. Noise Layer

The noise layer is consistent with the Stegastamp cover noise layer. Different from the design of HiDDeN model noise layer, the noise layer of HiDDeN mainly contains six different types of noise attacks, such as cropping, Gaussian noise, JPEG compression, etc. No suitable differentiable function has been designed to simulate the attack types generated in the screenshot process. Stegastamp’s noise layer is designed to resist the attacks generated in the screenshot process and printing process. In this work, we regard these attacks as the superposition of a series of attacks, including perspective deformation, motion and defocusing blur, color transformation, Gaussian noise and JPEG compression.

### 3.4. Moiré Network

In this paper, we first use the standard dataset in the field of moiré removal, which was proposed by Sun et al. [16]. This dataset contains 100,000 pairs of cover images and moiré images. We select 6000 pairs of images as the training set. On this basis, in order to improve the robustness of the watermarking algorithm in the archival image, we add some archival images into the dataset. The dataset image is shown in Figure 4. The shooting distance and angle are shown in Figure 5a. The distance of the phone from the monitor is about 20~30 cm, and the image obtained from the shooting needs to be further corrected for processing. Finally, 2000 pairs of moiré images are obtained as needed. Figure 5b shows the correct images. In order to improve the generalization ability of moiré network and to prevent the phenomenon of overfitting. Furthermore, the moiré network can accurately approximate the nonlinear relationship between the original image and the moiré image, and it saves the time cost of producing the dataset. The 2000 pairs of images we collected are augmented. Due to the lack of a standard moiré dataset, referring to paper [4,5], we decided to perform the geometric transformation on the existing images. The images are horizontally flipped, vertically flipped and scaled to expand the number of datasets. In the end, we obtained 6000 pairs of self-built moiré dataset.

In this paper, we propose a novel moiré network to improve the robustness of the algorithm by adding realistic moiré disturbance to the watermarked image. Unlike the cover Stegastamp, the noise layer use differentiable functions to simulate screenshot attacks. The moiré network does not apply the established functions to learn attacks, but it applies the learning characteristics of Convolutional Neural Network (CNN) to learn the moiré attacks in realistic scenes.

Figure 6 shows the network structure of the moiré network based on the U-net. The differently colored arrows in the figure represent different operations. Additionally, the two orange rectangles represent that they will be connected. The network architecture consists of three main parts: the encoding stage on the left, the decoding stage on the right and the hop connection. The encoding stage is used to down sample the cover image and extract high-level semantic information. The convolution in the encoding stage is a 3 × 3 convolution kernel with a step size of 1. The size of the convolution kernel for upsampling is 2 × 2. The decoding stage is used to decode high-level semantic features and generate the final moiré image after the moiré attack. Finally, the decoding stage changes the features into 3 channels by 1 × 1 convolution to obtain the noisy image after the moiré attack. The convolutional activation function is the Sigmoid function. All the other convolutions use 3 × 3 convolutional kernels and ReLU function. All padding methods in the network are of type SAME. The initializer of the convolution kernel is he_normal. Skip connection is applied to fuse the features of the two stages to make the result more accurate. Due to the weak of moiré in this dataset, we use iPhone 8, Hongmi K30 and Huawei hi nova9 mobile phones to collect the moiré images. Five hundred images are randomly selected from Div2k dataset [17] and displayed on the screen for shooting. The display used is HP N246v. The distance and angle of the mobile phone shooting are shown in Figure 7.

The distance between the mobile phone and the monitor is about 20~30 cm. An example of the image pair of the self-built moiré dataset is shown in Figure 7. In order to improve the generalization ability of moiré network, the collected 2000 pairs of images are augmented [18].

### 3.5. Decoder Construction

The decoder is used to extract the watermark from the moiré image. Some geometric attacks such as translation, scaling and rotation may be encountered in the transmission of moiré images. In order to balance the robustness of these geometric attacks and the watermark extraction accuracy, some preprocessing is needed before watermark extraction from the moiré image. Therefore, a spatial transformation network (STN) is used in the decoder, which can learn in an end-to-end manner without changing the loss function and improve the performance of the model. The final watermark information is obtained by Sigmoid function. The size of the convolution kernels of Conv_1~Conv_7 is 3 × 3. The activation function is ReLU. The filling method is same. The step size of Conv_1, Conv_3, Conv_5, Conv_6 and Conv_7 is 2, and the step size of other convolution operations is 1. The flatten operation is performed after Conv_7 to reduce the tensor dimension to 1 dimension. Then, the decoder performs two full concatenation operations to achieve the goal of successfully extracting the watermark information. The structure of the decoder is shown in Figure 8.

### 3.6. Model Training

DLM needs to select the appropriate loss function to guide network training. The loss function carries on the back propagation through the error between the predicted result and the ground truth. It also guides the learning of network parameters. In this paper, we choose different loss functions for different sub-networks. The mean square error (MSE) loss function and Learned Perceptual Image Patch Similarity (LPIPS) sensing loss function [19,20] are applied in the encoder, which are recorded as $L_{R1}$ and $L_{P}$. MSE loss function and LPIPS perceived loss function are shown in Equations (1) and (2), respectively.
(1)$$L_{R1}\left(I_{\mathrm{Cover}},I_{\mathrm{Stego}}\right)=\frac{1}{C\times H\times W}{\left\|I_{\mathrm{Cover}}-I_{\mathrm{Stego}}\right\|}_{2}^{2}$$
(2)$$L_{P}\left(I_{\mathrm{Cover}},I_{\mathrm{Stego}}\right)=\sum_{l}\frac{1}{H_{l}W_{l}}\sum_{h,w}{\left\|w_{l}\times \left({\hat{y}}^{l}-{\hat{y}}_{0}^{l}\right)\right\|}_{2}^{2}$$
where $I_{\mathrm{Cover}}$ is the cover image, and $I_{\mathrm{Stego}}$ is the watermarked image. ${\left\|\;\cdot \;\right\|}_{2}$ denotes summing the squares of each element and then calculating the square root of the sum. The l feature maps are extracted from the convolutional layer and normalized in channel dimension ${\hat{y}}^{l}\in R^{H_{l}\times W_{l}\times C_{l}}$ and ${\hat{y}}_{0}^{l}\in R^{H_{l}\times W_{l}\times C_{l}}$, respectively. We compute distance $L_{2}$ by scaling activation in the channel dimension by $W_{l}\in R^{C_{l}}$. Finally, the average distance in space and the sum in channel is calculated.

The loss function $L_{R2}$ of moiré network also uses MSE as shown in Equation (3)
(3)$$L_{R2}\left(I_{\mathrm{Cover}},I_{\mathrm{Screen}}\right)=\frac{1}{C\times H\times W}{\left\|I_{\mathrm{Cover}}-I_{\mathrm{Screen}}\right\|}_{2}^{2}$$
where $I_{\mathrm{Screen}}$ is the watermarked image after the moiré attack.

The decoder uses the cross entropy loss function $L_{M}$ [21], which is widely applied in classification tasks. The formula for $L_{M}$ is shown in Equation (4).
(4)$$L_{M}\left(M,M^{\prime }\right)=-\frac{1}{N}\sum_{i=1}^{N}\left[M_{i}\log\left({M^{\prime }}_{i}\right)+\left(1-M_{i}\right)\log\left(1-{M^{\prime }}_{i}\right)\right]$$
where M is the original watermark, $M^{\prime }$ is the extracted watermark, and N is the length of image watermark. The total loss function L of the model is shown in Equation (5)
(5)$$L=\lambda_{R1}L_{R1}+\lambda_{P}L_{P}+\lambda_{R2}L_{R2}+\lambda_{M}L_{M}$$
where $\lambda_{R1}$, $\lambda_{P}$, $\lambda_{R2}$ and $\lambda_{M}$ are the parameters of corresponding loss function. Finally, the encoder, the moiré network and the decoder are trained against each other, which make the model iterates continuously until convergence [22].

## 4. Experimental Results

### 4.1. Experimental Environment and Settings

The MIRFLICKR [23] dataset is used to evaluate the performance of the proposed algorithm ScreenNet. In the experiments, 20,000 images are selected for training, and the image size is adjusted to 400 × 400 pixels. The graphics card used is NVIDIA RTX 2080ti GPU training. The code is written in tensorflow framework. From the MIRFLICKR dataset, 2000 images which have no intersection with the training set are selected to form the verification set to evaluate the robustness under common noise attacks.

Watermarking information is a random sequence composed of 0 s and 1 s. Through experiments on different capacities of watermarking bits, the final choice is to embed 100 bits of watermarking information in the cover image. The distance similarity measures PSNR and SSIM are used to measure the quality of watermarked images. From Table 1, if the watermarking information bits increase, the value of PSNR and SSIM decrease. Consequently, the image quality will be poor, whereas the watermark capacity will be improved. Based on the comprehensive consideration of watermark capacity and image quality, 100 bits of watermarking information is selected to embed into the cover image. In this paper, the accuracy of extracting watermark information is used to evaluate the robustness of the proposed algorithm against a screenshot attack. The formula of the accuracy is shown in Equation (6).
(6)Accuracy=1-BER

### 4.2. Performance Comparison before and after Preprocessing

We compare the performance of the proposed ScreenNet algorithm before and after the preprocessing of the archival images. Figure 9 shows some cover archival images and the corresponding preprocessed archival images. We can notice that the text and color of the archival images can be identified from the preprocessed archival image. However, the archival images after preprocessing are similar to the archival images stored in reality.

### 4.3. Network Training Speed Comparison

We propose a novel ScreenNet algorithm based on a moiré network. Due to the addition of the moiré network in Stegastamp, the overall network architecture is more complex. So, the ScreenNet algorithm has a slower convergence speed compared to Stegastamp. Table 2 shows the training speed comparison between Stegastamp and ScreenNet. The training time of ScreenNet is about 2.3 h longer than Stegastamp. In the follow-up work, we need to standardize the data and select the appropriate learning rate to improve the training speed.

### 4.4. Network Training Speed Comparison

In this section, five common noise attacks are applied to evaluate the robustness of the proposed ScreenNet model. Figure 10 shows some visual images of watermarked images, noisy images and difference images. From Figure 10, the first row represents the watermarked images obtained by the proposed ScreenNet model. The second row shows the noisy images after the watermarked image is attacked. The third row represents the difference images between the watermarked image and the noisy image. The residual diagram $I_{re}$ between watermarked image and noisy images can be calculated from Equation (7) as follows:(7)$$I_{re}=\left|I_{en}-I_{no}\right|$$
where $I_{en}$ denotes the watermarked image and $I_{no}$ denotes the noisy image after the attack.

As shown in Table 3, five common noise attacks used in hidden are selected as the basis for evaluating the robustness, including Dropout (0.3), Cropout (0.3), Crop (0.035), Gaussian (2) and JPEG Compression (50). The numbers between brackets represent the noise intensity. From Table 3, we can notice that the accuracy of the proposed ScreenNet model is robust against Cropout, Dropout, Gaussian and JPEG compression. However, for crop attacks, the accuracy is only about 64%. The main reason is that the cover noise layer of Stegastamp does not consider the situation of crop attack in the training process, so the differentiable function is not applied to simulate this process. As a result, the watermark information is almost covered in the whole cover image, and a large amount of watermark information will be lost when the cropping attack is carried out.

### 4.5. Robustness Comparison of Screenshot at Different Distances

We evaluate the performance of the proposed ScreenNet model compared with HiDDeN and Stegastamp against screenshot attacks using different distances. Table 4 shows the anti-screenshot experiment.

The experimental results show that the HiDDeN model has poor robustness to screenshot attacks, and the accuracy of the watermark information extraction is in the range of 66~80%. This is because the noise layer of HiDDeN model only considers the common attacks such as cropping, JPEG compression, Gaussian blur, etc. However, it does not consider the screenshot attack. The Stegastamp model maintains a good watermark extraction accuracy under four different distances, and the accuracy rate is above 92%. This is mainly due to the application of various differentiable perturbations in the noise layer of Stegastamp to approach the screenshot attack. Compared with Stegastamp, the proposed ScreenNet model is more robust against screenshots. The proposed ScreenNet model also uses archival image dataset of moiré to add real disturbances. Therefore, the proposed ScreenNet has stronger antiscreenshot robustness. The average accuracy of watermark extraction with ScreenNet is higher than that of Stegastamp and HiDDeN models. Figure 11a shows the accuracy of screenshot attacks against different distances and angles. Furthermore, Figure 11a intuitively shows that the proposed ScreenNet model has a stronger robustness against screenshot under different distances.

### 4.6. Robustness Comparison of Screen Shots at Different Angles

In this experiment, we selected different perspective angles to take screenshots for the watermarked images. The relevant experimental data are shown in Table 5, and the graphical accuracy is shown in Figure 11b.

The experimental results show that the HiDDeN model has poor robustness to screenshot attacks under different perspective angles. The accuracy rate of the watermark information extraction is less than 60% for HiDDeN. The Stegastamp model also maintains good watermark extraction accuracy under different perspective angles, and its extraction accuracy of watermark information reaches about 80% of ScreenNet. Under different perspective angles, the accuracy of watermark extraction for the proposed ScreenNet model is slightly higher than the accuracy of Stegastamp and HiDDeN models. As ScreenNet applies the self-built moiré with more obvious moiré traces, the accuracy of watermark information extraction is improved. It can be noticed that the proposed ScreenNet algorithm is more robust to screenshot attacks.

## 5. Conclusions

In this paper, a robust screenshot watermarking auto-encoder ScreenNet based on Stegastamp model is proposed. It is mainly composed of the encoder, Stegastamp cover noise layer, moiré network and decoder. Differently from the previous CNN-based anti-screenshot robust watermarking algorithms, we no longer use differentiable functions to simulate the moiré noise in real scenes. Instead, we train the constructed moiré network by removing the standard moiré dataset and the self-built moiré dataset, so as to add real moiré noise to the cover image. Due to the lack of a suitable moiré dataset, a collection of large amount of data is presented. Finally, the encoder, Stegastamp noise layer, moiré network and decoder are trained against each other, and a robust anti-screenshot watermarking algorithm based on U-net is finally realized. The experimental results show that the watermarked image generated by the model has a strong robustness against screenshot attacks at different distances and perspective angles. Moreover, it has a high watermarking information extraction accuracy, which can effectively solve some practical problems. Additionally, archival images have important contents and a white background, which have many similarities with document images. Therefore, the algorithm in this paper is also well suited for document images.

In the future, we are planning to add an attention mechanism to the DLM to improve the performance of the model. Furthermore, we are aiming to improve the training speed of DLM by a reasonable way, such as adjusting unnecessary convolutional layers or well preprocessing the dataset.

## Figures and Tables

**Figure 1 entropy-25-00288-f001:**
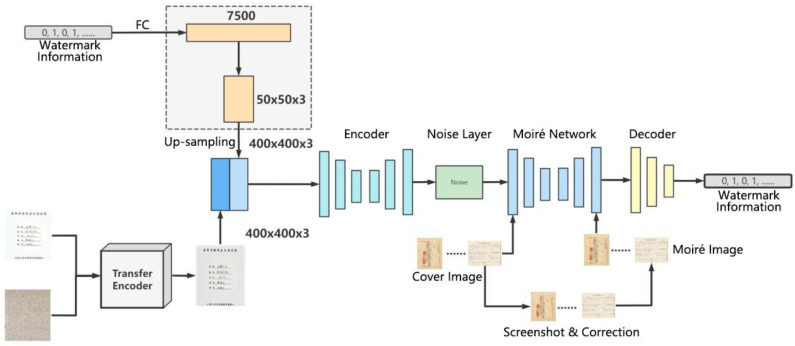
Flow Diagram of the Proposed Robust Watermarking Algorithm.

**Figure 2 entropy-25-00288-f002:**
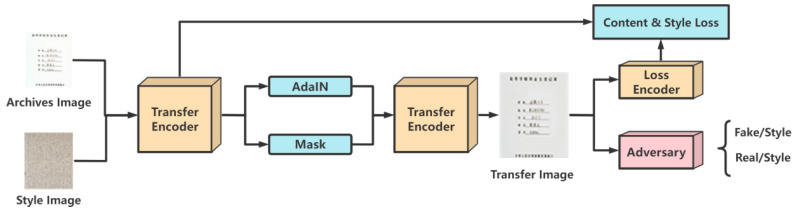
Archival image preprocessing based on VGG network.

**Figure 3 entropy-25-00288-f003:**
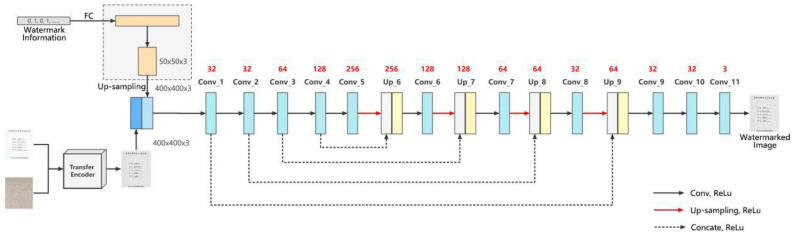
Encoder network structure.

**Figure 4 entropy-25-00288-f004:**
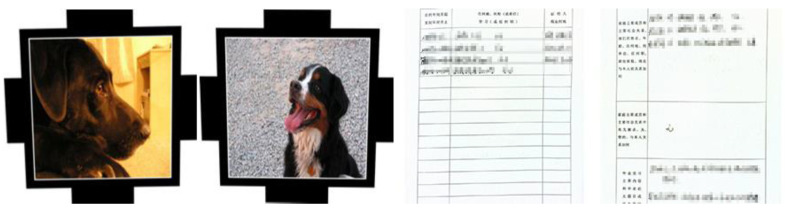
Example of our dataset image.

**Figure 5 entropy-25-00288-f005:**
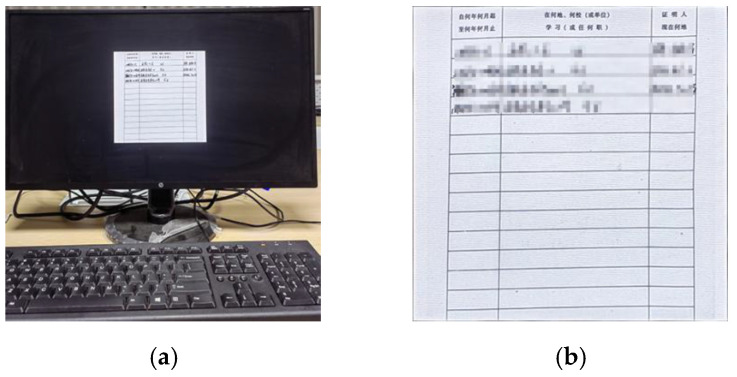
Collection and correction of moiré image. (**a**) Screen image; (**b**) Correct image.

**Figure 6 entropy-25-00288-f006:**
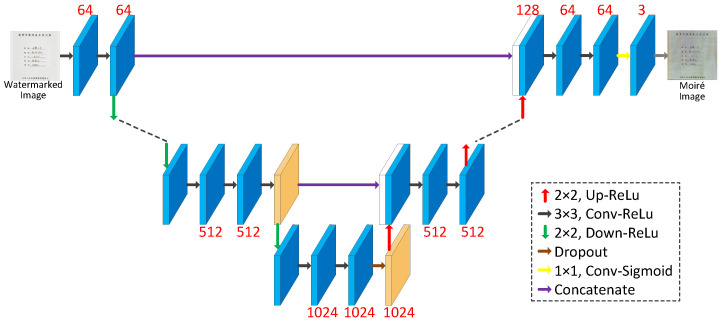
Structure of moiré network.

**Figure 7 entropy-25-00288-f007:**
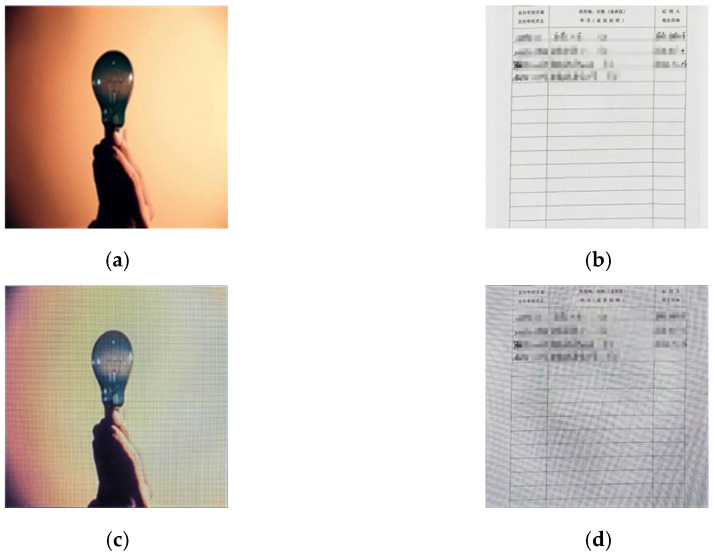
Example of self-built moiré image dataset. (**a**) A color image selected from the Div2k dataset. (**b**) The archival image. (**c**,**d**) The corrected images after screenshot, respectively.

**Figure 8 entropy-25-00288-f008:**
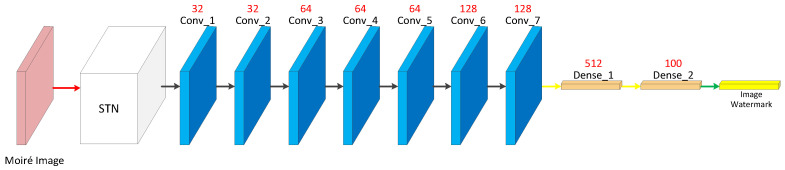
Decoder network structure.

**Figure 9 entropy-25-00288-f009:**

Archival images before and after preprocessing. (**a**,**d**,**g**,**j**) are archival images. (**b**,**e**,**h**,**k**) are four style images. (**c**,**f**,**i**,**l**) are their corresponding transfer images.

**Figure 10 entropy-25-00288-f010:**
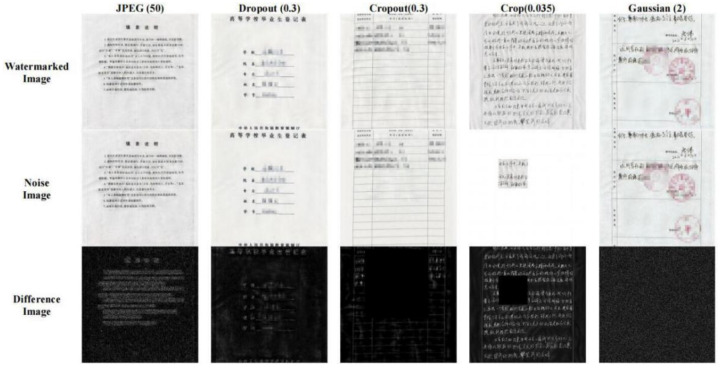
Watermarked archives image after noise attack and its residual image.

**Figure 11 entropy-25-00288-f011:**
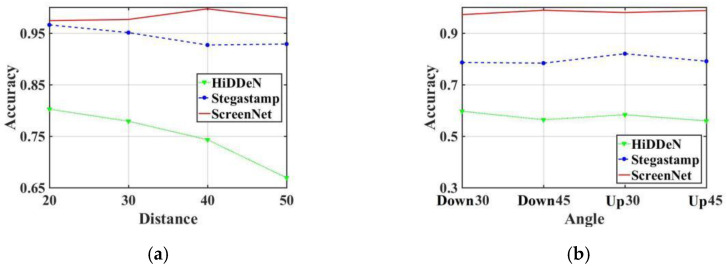
Screenshot attack accuracy against different distances and angles. (**a**) Distance; (**b**) Angle.

**Table 1 entropy-25-00288-t001:** PSNR and SSIM of watermarked image with different watermark capacity.

Evaluation Standards	50 bits	100 bits	150 bits	200 bits
PSNR	30.43	30.02	28.87	23.48
SSIM	0.923	0.919	0.845	0.808

**Table 2 entropy-25-00288-t002:** Comparison of network training time.

Method	Iteration Time	Per Iteration Time (s)	Total Time (min)
Stegastamp	140,000	0.185	432
ScreenNet	140,000	0.244	569

**Table 3 entropy-25-00288-t003:** Comparison of accuracy under different noise attacks.

Attack	Attack Range	Stegastamp	ScreenNet
JPEG Compression	20	0.9988	0.9931
Dropout	0.3	0.9594	0.9963
Cropout	0.3	0.8244	0.8673
Crop	0.05	0.6048	0.6405
Gaussian	2	0.9974	0.9823

**Table 4 entropy-25-00288-t004:** Accuracy comparison of screenshot attack at different distances.

Distance	20 cm	30 cm	40 cm	50 cm
Watermarked Image	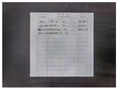	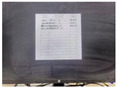	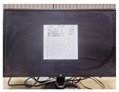	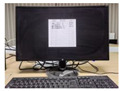
Corrected Image	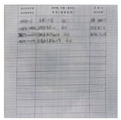	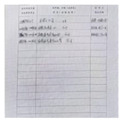	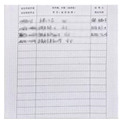	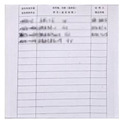
HiDDeN	0.8023	0.7789	0.7430	0.6692
Stegastamp	0.9658	0.9508	0.9267	0.9286
ScreenNet	0.9739	0.9763	0.9969	0.9790

**Table 5 entropy-25-00288-t005:** Accuracy comparison of screen shot attack under different angle.

Angle	30° down	45° down	30° up	45° up
Watermarked Image	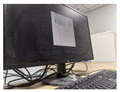	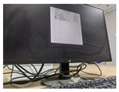	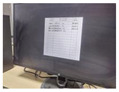	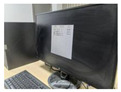
Corrected Image	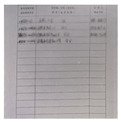	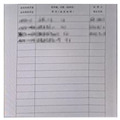	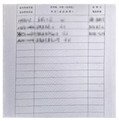	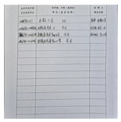
HiDDeN	0.5964	0.5634	0.5830	0.5591
Stegastamp	0.7857	0.7831	0.8197	0.7906
ScreenNet	0.9713	0.9883	0.9794	0.9873

## Data Availability

Not applicable.

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
