# Peer review of "Anti-Screenshot Watermarking Algorithm for Archival Image Based on Deep Learning Model"

_entropy, 2023, doi:10.3390/e25020288_

Round 1
Reviewer 1 Report
(1) In this paper, style transfer is used to pre-process for the archival images to improve the imperceptibility of the watermarking algorithm. Meanwhile, the Moiré network is used to generate more Moiré images for the noise layer of the deep learning model. The experiments show that the algorithm improves the robustness of the watermarking algorithm and is innovative.
(2) Some typos and grammar eros should be corrected. For example, there are a small number of English grammar problems. For example, “screenshotted” should be “screenshot” in Page 2; “stylization “ should be “style transfer” in Page 5. Please carefully check through the whole paper.
(3) Why use stylized network for background enhancement?
(4) Is the algorithm suitable for other images besides archival images?
Author Response
Point 1: In this paper, style transfer is used to pre-process for the archival images to improve the imperceptibility of the watermarking algorithm. Meanwhile, the Moiré network is used to generate more Moiré images for the noise layer of the deep learning model. The experiments show that the algorithm improves the robustness of the watermarking algorithm and is innovative.
Response 1: Thank you for your comments.
Point 2: Some typos and grammar eros should be corrected. For example, there are a small number of English grammar problems. For example, “screenshotted” should be “screenshot” in Page 2; “stylization “ should be “style transfer” in Page 5. Please carefully check through the whole paper.
Response 2: We have checked the full text for English grammar issues. In addition to the two mentioned issues, we also found other typos and grammar issues. For example, some abbreviations of proper nouns are not specified in detail when they are first applied. All the problems found have been corrected.
Point 3: Why use stylized network for background enhancement?
Response 3: We explained the reasons for using a stylized network at the beginning of Section 3.1.
Before:
An archival image has a large range of single background. However, the watermarking in a single background will be very conspicuous, which makes the algorithmic imperceptibility terrible. Therefore, the archival image needs to be preprocessed before being inserted into the model. In this work, we apply the style transfer algorithm to preprocess the archival image. This makes the archival image to add different colors and textures, which is more conducive to embed image watermark, improving both the imperceptibility and robustness of the proposed algorithm. The structure of the preprocessed archival image is shown in Figure 2.
Now:
Style transfer allows us to modify the style of an image to another style without changing the content of the image. This is great for archival images where content needs to be protected. More importantly, Style transfer makes the background of the archive image more suitable for embedding watermark information.An archival image has a large range of single background. However, the watermarking in a single background will be very conspicuous, which makes the algorithmic imperceptibility terrible. Therefore, the archival image needs to be preprocessed before being inserted into the model. In this work, we apply the style transfer algorithm to preprocess the archival image. This makes the archival image to add different colors and textures, which is more conducive to embed image watermark, improving both the imperceptibility and robustness of the proposed algorithm. The structure of the preprocessed archival image is shown in Figure 2.
Point 4: Is the algorithm suitable for other images besides archival images?
Response 4: We explain the problem at the end of the first paragraph of the Conclusion: And archival images have important contents and white background, which have many similarities with document images. Therefore, the algorithm in this paper is also well suited for document images.
Reviewer 2 Report
Regarding the redaction and the spelling:
- - There are some grammatical inconsistencies. Authors need to pay special attention to the concordance of numbs (e.g., In the abstract, use "one of the key problems", instead of "one of the key problem").
- Please, check that all acronyms are defined the first time they are introduced (e.g., CNN, HR).
- Some paragraphs are too large, I suggest splitting them depending on the main idea presented (e.g., 3rd paragraph of the introduction).
Regarding the content presentation
- Figure 8 is not referenced in the text.
- Some colors are used in Figure 6, but their purpose is not explained.
- It is recommended to cite some references in Section 2.1
- There are terms used in some equations that are not explained. It appears to be common knowledge, but I suggest their formal definition to avoid possible confusion among readers.
- The data given about reference 10 is poor.
About the content
- I would suggest a brief presentation of the term moiré for introductory readers in the related work section.
- The affirmation of the lack of existence of a standard moiré dataset should be addressed somehow, either by adding a citation or by referring to some cases.
Author Response
Regarding the redaction and the spelling:
- - There are some grammatical inconsistencies. Authors need to pay special attention to the concordance of numbs (e.g., In the abstract, use "one of the key problems", instead of "one of the key problem").
Response: We have checked the full text for English grammar issues. In addition to the two mentioned issues, we also found other typos and grammar issues. For example, “screenshotted” is modified to “screenshot”; “stylization “ is modified to “style transfer”. All the problems found have been corrected.
- Please, check that all acronyms are defined the first time they are introduced (e.g., CNN, HR).
Response: We checked all acronyms and made sure they were defined the first time they were used (e.g., Convolutional Neural Network CNN, Human Resources HR).
- Some paragraphs are too large, I suggest splitting them depending on the main idea presented (e.g., 3rd paragraph of the introduction).
Response: The 3rd paragraph of the introduction has been divided into three parts. The general introduction, directly using DLM as an encoder or decoder, and using DLM to assist in watermark extraction, respectively. We also added transitional statements at the beginning of parts 2 and 3.
Before:
However, the Stegastamp model is vulnerable for having visible noise on the watermarked image during the encoding process, which very unfavorable for the archival images with a single color background, so that it reduces the robustness and imperceptibility of the watermarking algorithm. Therefore, the performance of the StegaStamp model for archival images needs to be improved.
Now:
Stegastamp mainly improves the robustness of the image watermarking algorithm against Moiré attacks, which are interference streaks that appear on surface of the photographed object. For example, during screenshot, the image displayed on the shooting screen will appear Moiré pattern.
However, the Stegastamp model is vulnerable for having visible noise on the watermarked image during the encoding process, which very unfavorable for the archival images with a single color background, so that it reduces the robustness and imperceptibility of the watermarking algorithm. Some researchers have refined Stegastamp in the past two years, however, the performance of the model is still not well suited for archival images [8,9]. Therefore, the performance of the StegaStamp model for archival images needs to be improved.
Regarding the content presentation
- Figure 8 is not referenced in the text.
Response: We have cited Figure 8 at the end of Section 3.5: The structure of the decoder is shown in Figure 8.
- Some colors are used in Figure 6, but their purpose is not explained.
Response: We present the meaning of the different colors below in Figure 6: The different colored arrows in the figure represent different operations. And the two orange rectangles represent that they will be connected.
- It is recommended to cite some references in Section 2.1
Response: We have included two references on Stegastamp in Section 2.1.
- Wang K, Li L, Luo T, Chang C C (2020) Deep Neural Network Watermarking Based on Texture Analysis. Communications in Computer and Information Science, 558–569.
- Niu Y, Zhang J (2022) An image steganography method based on texture perception. Proceedings of the IEEE 2nd International Conference on Data Science and Computer Application.
- There are terms used in some equations that are not explained. It appears to be common knowledge, but I suggest their formal definition to avoid possible confusion among readers.
Response: The terms used in the equations have been explained in detail. For example, after Eq. (1,2): || . ||2 denotes summing the squares of each element, and then calculating the square root of the sum.
- The data given about reference 10 is poor.
Response: The original reference 10 now is reference 12. It's a electronic bulletin board online, and it’s uncomfortable to be added in this paper. Now we have deleted this reference.
About the content
- I would suggest a brief presentation of the term moiré for introductory readers in the related work section.
Response : We have included an introduction to the Moiré pattern in the second paragraph of Section 2.1:
Stegastamp mainly improves the robustness of the image watermarking algorithm against Moiré attacks, which are interference streaks that appear on surface of the photographed object. For example, during screenshot, the image displayed on the shooting screen will appear Moiré pattern.
- The affirmation of the lack of existence of a standard moiré dataset should be addressed somehow, either by adding a citation or by referring to some cases.
Response : We include a solution for the lack of a standard moiré dataset at the end of the first paragraph of Section 3.4: Due to the lack of a standard moiré dataset, referring to paper [4,5], we decided to perform the geometric transformation on the existing images.